# Glycan-Induced Protein Dynamics in Human Norovirus P Dimers Depend on Virus Strain and Deamidation Status

**DOI:** 10.3390/molecules26082125

**Published:** 2021-04-07

**Authors:** Jasmin Dülfer, Hao Yan, Maxim N. Brodmerkel, Robert Creutznacher, Alvaro Mallagaray, Thomas Peters, Carl Caleman, Erik G. Marklund, Charlotte Uetrecht

**Affiliations:** 1Heinrich Pette Institute, Leibniz Institute for Experimental Virology, 20251 Hamburg, Germany; jasmin.duelfer@leibniz-hpi.de (J.D.); hao.yan@leibniz-hpi.de (H.Y.); 2Department of Chemistry—BMC, Uppsala University, 75105 Uppsala, Sweden; maxim.brodmerkel@kemi.uu.se (M.N.B.); erik.marklund@kemi.uu.se (E.G.M.); 3Institute of Chemistry and Metabolomics, University of Lübeck, 23562 Lübeck, Germany; r.creutznacher@uni-luebeck.de (R.C.); alvaro.mallagaraydebenito@uni-luebeck.de (A.M.); thomas.peters@chemie.uni-luebeck.de (T.P.); 4Department of Physics and Astronomy, Uppsala University, 75105 Uppsala, Sweden; carl.caleman@physics.uu.se; 5Center for Free-Electron Laser Science, Deutsches Elektronen-Synchrotron, 22607 Hamburg, Germany; 6European XFEL GmbH, 22869 Schenefeld, Germany

**Keywords:** glycan interaction, norovirus capsid protein VP1, protruding domain, HDX-MS, native MS, hydrogen/deuterium exchange mass spectrometry

## Abstract

Noroviruses are the major cause of viral gastroenteritis and re-emerge worldwide every year, with GII.4 currently being the most frequent human genotype. The norovirus capsid protein VP1 is essential for host immune response. The P domain mediates cell attachment via histo blood-group antigens (HBGAs) in a strain-dependent manner but how these glycan-interactions actually relate to cell entry remains unclear. Here, hydrogen/deuterium exchange mass spectrometry (HDX-MS) is used to investigate glycan-induced protein dynamics in P dimers of different strains, which exhibit high structural similarity but different prevalence in humans. While the almost identical strains GII.4 Saga and GII.4 MI001 share glycan-induced dynamics, the dynamics differ in the emerging GII.17 Kawasaki 308 and rare GII.10 Vietnam 026 strain. The structural aspects of glycan binding to fully deamidated GII.4 P dimers have been investigated before. However, considering the high specificity and half-life of N373D under physiological conditions, large fractions of partially deamidated virions with potentially altered dynamics in their P domains are likely to occur. Therefore, we also examined glycan binding to partially deamidated GII.4 Saga and GII.4 MI001 P dimers. Such mixed species exhibit increased exposure to solvent in the P dimer upon glycan binding as opposed to pure wildtype. Furthermore, deamidated P dimers display increased flexibility and a monomeric subpopulation. Our results indicate that glycan binding induces strain-dependent structural dynamics, which are further altered by N373 deamidation, and hence hint at a complex role of deamidation in modulating glycan-mediated cell attachment in GII.4 strains.

## 1. Introduction

Noroviruses are non-enveloped single strand (+)sense RNA viruses of the *Caliciviridae* family that cause an estimated 20% of gastroenteritis cases worldwide [1]. The virus possesses an icosahedral capsid, consisting of dimers of the major capsid protein VP1. A minor capsid protein VP2 is located inside the icosahedral shell. Based on its VP1 sequence, ten genogroups have been categorized, of which GI, II, IV, VIII and IX can infect humans [2]. Noroviruses of genogroup II (GII), especially genotype GII.4, dominated outbreaks in the last two decades [1]. With the emergence of new strains, e.g., GII.17 in Asia, it is an open question as to whether GII.4 will be displaced [3,4] or resurgent [5].

VP1 is divided into the inner shell (S) domain and the outward-facing protruding (P) domain [6]. The P domain is further subdivided into P1 and P2 subdomains, with P2 being essential for host immune response and binding to histo blood-group antigens (HBGAs) for cell attachment in a strain-dependent manner [7]. The importance of interactions with HBGAs for host cell attachment has been shown in several studies [8], but how these interactions actually mediate cell entry remains unclear. While a comparable small animal model or robust cell culture system for human norovirus is still unavailable, P dimers and virus-like particles (VLPs) are considered a valid model for glycan-binding studies [6,9,10]. So far, several biophysical techniques including nuclear magnetic resonance (NMR), X-ray crystallography, native mass spectrometry (native MS) and hydrogen-deuterium exchange mass spectrometry (HDX-MS) have been applied to characterize binding of P dimers and VLPs to HBGAs and other glycans [11,12,13,14,15,16,17,18,19,20]. These studies revealed that glycan preferences and binding affinities are strongly genotype- and strain-dependent. For instance, crystallization studies showed that two fucose binding pockets on the top of the P dimer are highly conserved among different strains [8,21], while in the GII.10 Vietnam 026 strain, two additional binding sites located in the P2 cleft are occupied at high fucose concentrations [14] (Figure 1C,D). For GII.4 P dimers, recent chemical shift perturbation titrations demonstrate the presence of two independent HBGA binding sites [15,22] suggesting that the four fucose binding sites shown for GII.10 Vietnam P dimers are an exception.

Recently, NMR measurements identified a spontaneous deamidation of N373 with subsequent formation of an iso-aspartate (iD) in GII.4 Saga P dimers that strongly attenuates glycan binding. This deamidation appears to be site specific and occurs in GII.4 MI001 P dimers as well, whereas it is absent in GII.10 Vietnam 026 and GII.17 Kawasaki 308 P dimers, which carry a Gln or Asp at the equivalent position [15]. HDX-MS measurements showed binding site protection only in the wildtype P dimer, confirming the loss of HBGA B trisaccharide binding in deamidated P dimers, which correlated with increased flexibility in the P2 domain compared to the wildtype P dimer.

HDX-MS measures the exchange of protein backbone hydrogens to deuterium in solution. As this exchange strongly depends on solvent accessibility and hydrogen bonding patterns, the method can provide information about regions involved in ligand binding as well as changes in protein dynamics in solution [23]. This makes it a valuable technique for identification of glycan induced structural dynamics in different strains as well as elucidation of altered protein dynamics in deamidated P dimers. While P dimers across strains are structurally highly similar, their glycan binding behavior and infectivity is highly variable, leading to the hypothesis that varying structural dynamics are linked to these different profiles.

Therefore, we set out to examine whether glycan binding or deamidation can induce distinct structural dynamics in P dimers that could be important for glycan-mediated cell attachment. We specifically investigated binding of HBGA B trisaccharide and l-fucose to P dimers of GII.4 Saga, GII.4 MI001, GII.17 Kawasaki 308 and GII.10 Vietnam 026, significantly extending previous NMR studies [15,22]. GII.4 MI001 infects humans and mice [24] and has been chosen as comparison to the almost identical strain GII.4 Saga. GII.17 Kawasaki 308 is an emerging strain and the less abundant GII.10 Vietnam 026 is capable of binding four fucose molecules per P dimer. The structural aspects of glycan binding to fully deamidated GII.4 P dimers have been investigated before [15]. However, considering the high specificity and short half-life of N373D under physiological conditions [15], large fractions of partially deamidated virions with potentially altered dynamics in their P domains are likely to occur. Therefore, we also examined glycan binding to partially deamidated GII.4 Saga and GII.4 MI001 P dimers.

Our targets share varying amounts of sequence identity with the GII.4 Saga P dimer, but are highly similar on the structural level (Figure 1A,B), with largest differences in the loop regions of the P2 domain. Our results reveal protection of the canonical binding site in all wildtype P dimers. While glycan binding behavior in GII.4 Saga and GII.4 MI001 strains is identical, distinct glycan induced dynamics are observed in GII.17 Kawasaki 308 and GII.10 Vietnam 026. Furthermore, all strains apart from GII.4 Saga present a second P domain species that is highly protected from HDX. In partially deamidated GII.4 P dimers, fucose binding leads to different structural dynamics than in pure wildtype or fully deamidated samples, hinting at a potential biological function. Moreover, molecular dynamics (MD) simulations with an aggregated simulation length of 14 μs are employed to dissect the origin of observed differences.

## 2. Results

### 2.1. P Dimer Quality Control by Peptic Digest and Native MS

To verify the deamidation status, P dimer samples were subjected to peptic cleavage followed by LC-MS for peptide identification (Appendix A). No deamidated peptides were identified for GII.10 Vietnam and GII.17 Kawasaki after prolonged storage at 5 °C (Vietnam: 1 year and 4 months, Kawasaki: 1 year). For GII.4 MI001 P dimers stored for 1 year at pH 7.3 and 5 °C, a fraction of approximately 64% was deamidated at N373. Based on this the ratio of purely native (NN) to half deamidated (iDN) to fully deamidated (iDiD) P dimers is statistically predicted as 13:46:41%. Furthermore, a minor fraction was deamidated at N239 and N448, respectively. For GII.4 MI001 P dimers stored at pH 4.9 for 5 months at 5 °C, no deamidation of N373 was observed. Only a small fraction (<10%) of deamidated N448 was identified. GII.4 Saga P dimers stored for more than 2 years at pH 7.3 at 5 °C were approximately 88% deamidated at N373 leading to a ratio of NN:iDN:iDiD of 1.5:21:77.5% (Appendix A). In GII.4 datasets, the low intensity of peptides covering the N373 deamidation site precluded HDX analysis, while the low fraction of deamidated peptides at the other deamidation sites were deliberately excluded.

Prior to HDX-MS analysis P dimers were subjected to native MS for quality control. Furthermore, ion exchange separated wildtype (NN) and fully deamidated (iDiD) GII.4 Saga P dimers were measured for comparison. GII.17 Kawasaki, GII.10 Vietnam, wildtype GII.4 MI001 and wildtype GII.4 Saga P domains showed dimers with the expected molecular masses, apart from a small fraction of tetramers likely formed during the electrospray ionization (ESI) process (Appendix A). Interestingly, both deamidated GII.4 P domains were also present as monomers. Increased monomer fractions correlate with the extent of N373 deamidation: 16% monomers are detected for the 64% deamidated GII.4 MI001 sample and 32% monomers are found for the 100% deamidated GII.4 Saga sample (Appendix A). As deamidation rates in these strains are identical [22] this suggests that monomers are a result of iDiD P dimer dissociation, while iDN species are still primarily dimeric.

### 2.2. Bimodality of Deuterated Peak Distributions Indicates a Lower Deuterated Protein Subpopulation

Based on glycan binding data for GII.4 Saga P dimers [15,22], we wanted to expand our HDX MS experiments to P dimers of other human norovirus strains to analyze possible strain specific differences in the structural response to glycan binding. Therefore, we incubated GII.4, GII.10 and GII.17 P dimers with 10 mM HBGA B trisaccharide or 100 mM fucose at pH 7 and measured differences between the unbound and ligand-bound state using HDX-MS. That such high glycan concentrations itself do not pose a problem had been tested previously using 100 mM galactose as negative control [15]. During inspection of the glycan binding data, we observed that deuterated spectra of some peptides had a bimodal character with a first low intense, low deuterated and a second high intense, higher deuterated peak distribution (Figure 2A). Notably, bimodality was observed in all but the GII.4 Saga datasets. These bimodal peak distributions can have many causes, e.g., two distinct protein conformations, conformational rearrangements that lead to EX1 exchange kinetics, insufficient ligand saturation or peptide carryover from the analytical or, most often, protease column [25,26].

To rule out that the effects are solely induced by carryover, an additional dataset with randomized sample order and additional washing of the pepsin column between sample injections was measured for GII.4 MI001 wildtype (wt, N373) P dimers incubated with fucose (Figure 2C, experiment 6). This dataset still showed bimodality, but reduced relative intensity of the first peak distribution compared to experiment 5 without the wash step. This indicates that in all experiments without the wash step (experiments 1–5) varying amounts of peptide carryover added onto the lower deuterated signal of the more compact subpopulation. Moreover, bimodality is also observed in absence of any ligand, proving that undersaturated binding sites are not the origin of bimodality. Additionally, ligand concentrations were chosen to provide high and comparable saturation of binding sites.

For all three strains, bimodality is almost exclusively present in the P2 domain and in the lower part of the P1 domain (Figure 2B). Peptides that are affected by glycan binding often show bimodal peak distributions as well; therefore, it was necessary to manually analyze deuteration differences in these regions by binomial fitting of the individual peak distributions (an example analysis can be seen in Appendix A). Notably, peptides covering the canonical binding site (Table 1) are unimodal in GII.17 Kawasaki and GII.10 Vietnam and only show slight bimodality in GII.4 MI001 in the presence of fucose.

Relative intensities of the individual peak distributions are constant over the deuteration time (Appendix A) and highly similar for peptides within the same protein, which lead us to the assumption that the P dimer is present in two distinct subpopulations: a compact (lower deuterated) and a more flexible (higher deuterated) one. This is further supported by increased deuteration in both populations over time, albeit to a small extent for the low deuterated component (Appendix A). The relative intensity ratios of the peak distributions vary between experiments, but can still be compared within a certain experiment (Figure 2C). Depending on the strain and the experiment, the relative intensity of the first peak distribution in the unbound P dimer varies between 7 and 17%. Incubation with HBGA B trisaccharide has no (GII.4, GII.17) or just a low (GII.10) effect on the relative intensity of the first peak distribution. Presence of fucose, in contrast, significantly increases the relative intensity of the first distribution (i.e., compact subpopulation) in all strains.

### 2.3. Analysis of Glycan Induced Changes in P Dimers of Different Strains by HDX-MS

Most of the deuteration changes in presence of HBGA B trisaccharide or fucose are detected in peptides that show bimodality. This commonly causes falsely low deuteration differences in centroid analysis, so individual binomial fitting of the two peak distributions was performed for some representative peptides in these regions to validate the observed deuteration differences in the main (second) peak distribution. Woods’ plots indicating significant deuteration differences and regions of bimodality for all datasets can be found in the Appendix A. Representative uptake plots for all regions with significant deuteration differences after validation by binomial fitting are presented in the Appendix A. HBGA B trisaccharide affinities are almost identical for GII.4 Saga and GII.4 MI001 P dimers [22]. Assuming transferability to P dimers of other strains, binding pocket occupancy can be estimated based on K_d_ values measured by NMR (GII.4 Saga (NN): K_d_ = 5.5 mM for HBGA B trisaccharide and K_d_ = 22 mM for fucose) [15]. Binding of HBGA B trisaccharide in our setup would hereby correspond to 95% binding site occupancy during equilibration with ligand (98% for fucose) and 65% during deuterium labeling (82% for fucose). As expected, binding of HBGA B trisaccharide and fucose induces changes in P dimers of all three strains, primarily in the P2 domain, indicating occupation of the glycan binding pocket (Figure 3 and Table 1).

Protected regions in wildtype GII.4 MI001 P dimers are highly similar to GII.4 Saga P dimers [15] for both glycans (canonical binding site G443, Y444 and residues 283–303) indicating dataset validity. In addition, protection of a β-sheet region in the top cleft of the P2 domain (residues 333–353) can be detected in GII.4 MI001 in presence of HBGA B trisaccharide. Chemical shift perturbations (CSP) in this region could also be seen in NMR experiments with GII.4 Saga P dimers in presence of glycans [15,22]. Overall, protected regions in GII.4 MI001 match with regions showing CSPs in GII.4 Saga NMR data, suggesting that both strains respond similarly to glycan binding.

Protection of residues 333–353 in the P2 domain can be seen in all strains. For GII.17 Kawasaki P dimers, significant protection in presence of HBGA B trisaccharide is only present in this specific region. When incubated with fucose, additional protection of the canonical glycan binding site (G443, Y444) and residues 269–286, located in the protein center below the P2 domain, can be detected (Figure 3A). GII.10 Vietnam P dimers also show protection in their canonical binding site (G451, Y452) and the β-sheet region in the binding cleft of the P2 domain (residues 337–364 and 379–399) is protected as well (Figure 3D). All differences depicted in Figure 3 can only be seen in the second, highly deuterated peak distribution. The lowly deuterated peak distribution showed no significant differences between the unbound and the glycan-bound state in any of the strains indicating that either only the highly deuterated species can bind glycans or labeling time was too short to detect deuteration differences in already strongly protected regions.

### 2.4. Influence of N373 Deamidation on Dynamics and Glycan Binding of GII.4 P Dimers

To study the influence of partial vs. complete deamidation of N373 on glycan binding [15], HDX-MS experiments with partially deamidated GII.4 MI001 and GII.4 Saga P dimer samples in the presence of 100 mM fucose were performed. Strikingly, protection of the canonical fucose binding site (G443, Y444) could be detected in both strains (Figure 4A,C,D). This shows that fucose binding is still possible in partially deamidated (iDN) or even fully deamidated (iDiD) P dimers at the given concentration, even though binding is attenuated compared to the N373 wildtype [15]. Occupation of the canonical binding sites has been seen in crystal structures of deamidated GII.4 Saga P dimers at elevated concentrations of 600 mM fucose, but binding interactions were slightly different from wildtype [15]. In contrast to wildtype GII.4 MI001 P dimer, no other region was protected from HDX under fucose treatment, but increased deuteration in the main (second) peak distribution was observed in the P2 domain of both GII.4 strains, suggesting a more exposed conformation (Figure 4D). Interestingly, residues 335–362, which are protected in the wildtype proteins of all strains, show increased deuteration in partially deamidated GII.4 P dimers upon glycan binding. As we have a mixture of wildtype and deamidated P domains in the sample, the mass shift we see in the deuterated spectra reflects the average of all components, unbound and fucose-bound NN, iDN and iDiD P dimers, which cannot be discriminated. However, binding probability calculations can give a hint, which species contribute most to the observed increase in deuteration in presence of fucose. Considering the fractions of wildtype and deamidated P domains and their different K_d_s for fucose binding [15], in GII.4 MI001 only 17% of binding events occur in pure wildtype NN dimers, 54% in half-deamidated iDN dimers and 29% in fully deamidated iDiD dimers. For GII.4 Saga even more binding events take place in fully deamidated P dimers (2:31:67% for NN, iDN; iDiD), clearly showing that the detected increased deuteration is caused by fucose binding to at least half-deamidated P dimers. This suggests that fucose binding results in different dynamics in the wildtype and partially deamidated protein.

In GII.4 MI001 regions with bimodal peak distributions, relative intensities of both distributions are similar to the ones in the wildtype protein. However, interaction with fucose in the partially deamidated GII.4 MI001 P dimer does not lead to a significant increase in the relative intensity of the lower deuterated peak distribution, as seen in the wildtype protein (Figure 4F,G). Slight bimodality is also present in peptides covering the canonical fucose binding site in the fucose bound state but relative intensities are similar to the ones observed for the wildtype protein.

A comparison of FD normalized deuteration levels for wildtype and deamidated GII.4 MI001 P dimers without glycans reveals increased deuterium incorporation in large parts of the P2 domain, as well as the P1 domain (Figure 4B). Highest deuteration differences (∆D > 1 Da) are detected for residues 335–432 located in the P2 β-sheet cleft (Figure 4E). This is in line with the increased flexibility in the P2 domain of the deamidated GII.4 Saga P dimer [15] (Figure 4B); however, in GII.4 MI001 this effect is propagated into regions more distant from the glycan binding pocket and deamidation site. The increased dynamics could weaken the dimer interfaces largely formed by the P2 domain and therefore explain the dissociation into monomers in the deamidated protein. Additionally, monomers will most probably also experience slightly higher HDX because of missing dimer interactions in the P1 domain and the P2 β-sheets [28] that will add to the observed increase in deuteration compared to the exclusively dimeric wildtype protein.

### 2.5. MD Simulations

MD simulations were utilized to further investigate the norovirus P dimer strains. The RMSD relative to the starting structure post-equilibration were calculated in order to estimate protein dynamics during MD simulations. RMSF calculations were employed to examine fluctuations throughout the simulated time frame and highlight alterations in flexible regions of the different protein chains. As support for the RMSF data, we calculated the A_sas_ of the P dimers during the simulation with respect to their crystal structure, providing an understanding of an increase or decrease of the surface area of each individual residue.

The RMSD for the four norovirus P dimer strains are reported in Appendix A, in which the simulated GII.4 Saga and MI001 P dimers reached a value of 1.5 Å after 100 ns. GII.10 Vietnam P dimers show a maximum deviation around 90 ns at 2 Å, decreasing to 1.8 Å after 100 ns. GII.17 Kawasaki P dimers demonstrate a still slightly increasing trend at the end of the simulation, indicating that this system has not yet fully adapted to the solution environment. The RMSFs and A_sas_ relative to the respective crystal structures reveal differences in protein chain flexibility of the norovirus strains, as depicted in Figure 5 and Appendix A. The sequences were aligned for a better comparison. Hence, resulting gaps in the individual RMSF graphs are due to missing residues at that specific position. Least stability is introduced for the GII.4 Saga strain, as the RMSF values suggest only a limited increase in fluctuation during the 100 ns of simulation. GII.4 MI001 and GII.17 Kawasaki follow a similar trend. In contrast, GII.10 Vietnam P dimers show overall higher flexibility compared to the other strains, in particular a peak around residue 350 in the P2 domain. Similar trends can be observed for the A_sas_ graphs depicted in Appendix A. GII.10 Vietnam demonstrates the highest area values, which support the peaks observed in the RMSF graph in Figure 5A. The various P dimer structures were overlaid in the PyMOL software, where areas of interest were imaged in order to further explore differences of the protein crystal structures (Appendix A).

Investigating GII.4 Saga P dimers complexed by fucose and the potential role of deamidation, the data revealed minimal difference between the RMSD values of said systems. RMSDs for the NN, iDN and iDiD G.II 4 Saga dimers show a similar trend, reaching a value between 1.6 and 1.75 Å after 100 ns of simulation (Appendix A). RMSF and A_sas_ calculations to investigate the role of deamidation in the GII.4 Saga strains show that the individual graphs follow a similar trend, suggesting only limited influence of the deamidation on the overall P dimer structure (Appendix A) which is in line with previous crystallography data [15].

## 3. Discussion

In this study, we address the differences in structural responses to glycan binding of norovirus P dimers of the Asian epidemic strain GII.17 Kawasaki 308, the rarely detected strain GII.10 Vietnam 026 and the GII.4 MI001 strain, which belongs to the highly epidemic GII.4 genotype and has been shown to infect mice as well [24].

### 3.1. Putative Origin of Bimodal Peak Distributions

In our glycan-binding data, we observe the presence of bimodal peak distributions in a large variety of peptides located in the P2 domain and the lower part of the P1 domain in all strains, but not in the previously analyzed GII.4 Saga. The intensity of the lower deuterated peak distribution was between 7% and 17% and relative intensities of both distributions remained constant over time. This observation points towards two distinct protein subpopulations [29] that experience a different level of HDX over the whole exchange period. The low deuteration of the first peak distribution suggests the presence of a more compact subpopulation that is shielded from HDX. The P domain can form larger oligomers of different stoichiometry, up to whole 24-mer P particles, depending on the protein concentration [30,31]. These P particles can bind HBGAs and are even suspected to interact with them in the same way as VLPs [30,32]. They have been used in mutagenesis studies [33,34] and are discussed as potential vaccine platforms [35]. P oligomers form contacts through interactions in the lower part of the P1 domain of each P dimer [30], which could explain the reduced deuteration in this area. Closer inspection of the cryo EM structure [30] also suggests more contacts between the P2 domains compared to free P dimer. Importantly, the absence of bimodality in GII.4 Saga P dimers implies that this strain would have a different ability to form P oligomers than the closely related GII.4 MI001 strain.

In our data, there is a clear increase of the compact subpopulation in presence of 100 mM fucose indicating a glycan induced transition between the two subpopulations. Interestingly, this could mean that interaction with glycans supports otherwise less pronounced P particle formation [32]. It has to be noted that we did not observe P particle oligomers in size-exclusion chromatography as well as native MS of a 4.5 µM P dimer solution and our protein constructs lack the C-terminal arginine cluster that has been shown to be important for efficient P particle formation [36,37]. However, the <15% of compact monomers in absence of glycans would amount to around 1% of 24-mer P particle relative to the total signal intensity. Additionally, split up into many charge states in native MS could result in drop below detection limit. In contrast, fractions of structural variants of less than 5% can be detected in a properly conducted HDX-MS experiment [38]. Overall, the presence of P particles could be an explanation for the bimodal behavior of deuterated peak distributions, but cannot be undoubtedly proven with the experiments applied. Importantly, the occurring lower deuterated subpopulation needs to be treated separately to not obscure the data of the main subpopulation, which reflects the biologically relevant P dimer fraction.

### 3.2. Glycan Binding in Different Strains

When comparing the deuteration levels of the higher deuterated (second) peak distribution originating from P dimers, we could detect differences in glycan-induced protein dynamics in different strains. So far, glycan-binding CSP data from NMR is only available for GII.4 strains [15,22]. Hence, our HDX-MS data extends previous studies providing new insights into protein dynamics of different strains. For GII.4 MI001, regions protected from H/D exchange were almost identical to the earlier investigated GII.4 Saga P dimer (canonical binding site G443, Y444 and residues 283–303) [15], apart from additional protection in the upper P2 binding cleft (residues 333–353). Involvement of this region has been seen in NMR data of GII.4 Saga P dimers as well [15]. Furthermore, a recent NMR study suggests identical glycan binding behavior of both GII.4 strains [22]. The same study also shows that MNV P dimers do not bind HBGAs, underscoring that infectivity of GII.4 MI001 in mice cannot be explained by different glycan-induced dynamics between GII.4 Saga and MI001 in line with our observations.

GII.17 Kawasaki P dimer crystal structures with fucose and HBGA A trisaccharide show backbone interactions in T348 and G443 and side chain interactions in R349, D378 and Y444 [13,22]. When incubated with HBGA B trisaccharide and fucose, protection from HDX is observed for residues 333–353 corresponding to interactions with T348 and R349. In the presence of 100 mM fucose, the canonical binding site (G443, Y444) is protected, as well as residues 269–286, which cannot be explained by the known interactions from the crystal structures. This region is located below the glycan binding cleft in the protein center, so protection from HDX could rather be the result of a long-distance structural change than of direct interaction with fucose. It would be interesting to see how long-distance structural changes would further propagate into the S domain in VLPs and if they would influence the dynamic P domain lift off from the S domain that has been seen for different norovirus strains [39,40,41].

For the GII.10 Vietnam strain, binding of two HBGA B trisaccharide molecules and up to four fucose molecules has been seen in crystal structures [12,14]. Compared to GII.4 MI001 and GII.17 Kawasaki, we see protection in more protein areas for both HBGA B trisaccharide and fucose, which mainly corresponds to the known glycan interactions summarized in Table 1. Due to close proximity of interacting amino acids in fucose binding sites 1/2 and 3/4 we cannot distinguish these binding sites in HDX data at peptide resolution, but occupation of all four binding sites is likely at the given concentration [14]. Interestingly, we see a protection of several residue stretches that cannot be explained by known glycan interactions. Residues 311–336 belong to an unstructured region below the P2 binding cleft. In presence of HBGA B trisaccharide, protection of the aforementioned residues is not present under the chosen conditions. A possible explanation could be that these changes in dynamics are triggered by occupation of binding sites 3 and 4 in the P2 cleft, which so far has not been seen for HBGA B trisaccharide at similar concentrations [14]. HBGA B trisaccharide binding is mainly mediated by the fucose residue, with an additional interaction of galactose with G451 and some water mediated interactions [12]. In our data we detect protection of residues 483–496 on the bottom of the P dimer in addition, which could be a long-range effect not triggered by fucose alone.

Taken together, P dimers of all investigated strains showed protection of the upper P2 binding cleft (residues 333–353) underscoring the importance of this region for glycan binding as well as the consistence between datasets. Protection of the canonical glycan binding site (G443, Y444 for GII.4 and GII.17; G451, Y452 for GII.10) was also detected in all strains and for all glycans apart from HBGA B trisaccharide binding with GII.17 Kawasaki P dimers. HBGA B trisaccharide could have a lower binding affinity in GII.17 Kawasaki compared to the other strains that leads to smaller deuteration changes that are below the detection limit in the current experimental setup. We also noticed that the GII.17 Kawasaki datasets have a higher back exchange (D/H) than the other datasets so that small glycan induced deuteration changes are more likely to be lost during the measurement. GII.4 P dimers show protection of residues 285–298, which is absent in GII.17 and GII.10 P dimers. Interestingly, P dimers of the more prevalent strains GII.4 and GII.17 [3,4] show less changes in HDX upon glycan binding compared to GII.10 Vietnam, which is rarely detected in patients [12].

The RMSD plots for the four investigated P dimer strains without ligand reveal minimal differences between all systems (Appendix A). Whilst GII.4 Saga and MI001 trends reach a plateau, one can observe a still increasing trend in GII.17 Kawasaki. This indicates that this system has not yet reached a stable conformation. The GII.10 Vietnam shows a decrease towards the end of the simulation, indicating that this system just adapted to the environment and obtained a stable structure.

For the MD simulations, we were interested in the dynamics in absence of fucose and we observed an increase in flexibility throughout the peptide chain, accompanied with changes of the A_sas_ (Figure 5). The most prominent difference between the strains is a high peak around the glycan binding site near residue 350 exclusively in GII.10 Vietnam P dimers. GII.10 Vietnam has a longer loop around residue 350, which could explain the higher flexibility in absence of fucose (Figure 5A,B). On further investigation, however, when complexed by fucose in the crystal structure, this loop adopts a short helical structure (Figure 6A), forming a pocket shielding nearby residues from deuterium exchange, as observed in the HDX-MS experiment (Figure 6C). The crystal structures of the other strains have more unstructured loops. In our simulations of ligand-free GII.10 Vietnam P dimers, the former structured loop becomes flexible and unstructured, as seen by the high RMSF values of up to 7.5 Å and evident from snapshots taken from the MD trajectory (Figure 6B). This is further supported by the increase of area accessible by the solvent (Figure 5B). As such, for GII.10 Vietnam, the binding of fucose or other ligands promotes a structural rearrangement of the glycan binding site near residue 350 (Figure 6A) not seen in the other strains.

Around residue 250 and 300 the GII.10 Vietnam strain presents increased flexibility compared to GII.4 Saga, GII.4 MI001 and GII.17 Kawasaki. Near residue 424, a smaller peak can be observed for all four strains (Figure 5). This could be a result of the individual chain orientations and the different sequence alignment one can find in these areas (Appendix A–C). The residues that are part of this area of interest seem to form smaller loops on the surface of the protein, which is likely the reason of the recorded high flexibility.

### 3.3. The Role of N373 Deamidation

For GII.4 Saga P dimers, it was observed earlier that spontaneous transformation of N373 into iso-aspartate (iD) attenuates glycan binding in fully deamidated iDiD P dimers. Deamidation is site-specific and happens over a timescale of 1–2 days at pH 7.3 and 37 °C correlating with the length of the infection cycle [15]. The high specificity of this deamidation under infection conditions suggested that this process could occur as well in vivo; however, the biological relevance for infection remained unclear.

To test if this effect can also be found in the closely related GII.4 MI001 strain, we performed HDX-MS of fucose binding to a spontaneously deamidated P dimer sample, containing mixed populations of NN, iDN and iDiD dimers. Considering the deamidation rate under physiological conditions, such a mixture is more likely to be found in a natural infection context, than the earlier investigated pure iDiD P dimer [15]. For comparison, we also measured fucose binding to a partially deamidated GII.4 Saga P dimer, which contained an even higher fraction of fully deamidated iDiD P dimers. Strikingly, protection of the canonical glycan binding site from H/D exchange could still be detected in both isolates under fucose treatment, mainly corresponding to binding to iDN and iDiD P dimers. Additionally, the P2 cleft was more exposed in the partially deamidated sample under fucose treatment, in contrast to the protection observed in the wildtype NN P dimer. This indicates that under natural deamidation conditions, glycan binding at the canonical binding site still happens, but induces different dynamics than in the purely wildtype P dimer. The exposure of the P2 cleft suggests increased flexibility after glycan binding in this area, which could be required to interact with other factors or the until now unknown receptor. As such an increase in deuteration is not present in wildtype NN P dimers, this effect must be caused either by direct binding to deamidated P domains in iDN or iDiD dimers or by binding to wildtype P domains in iDN dimers, whose overall dynamics are altered by the influence of the neighboring deamidated monomer. RMSD, RMSF and A_sas_ calculation for the GII.4 NN, iDN and iDiD Saga P dimers show no striking differences when compared to each other and follow a similar trend (Appendix A). The fact that our MD simulations were unable to detect differences between deamidated and non-deamidated P dimers suggests that any differences in dynamics are manifested on timescales longer than a few 100 ns. Neither protection of binding sites nor increased deuteration in P2 has been seen in previous HDX-MS measurements of fully deamidated GII.4 Saga P dimer with 10 mM HBGA B trisaccharide under nearly identical conditions [15]. A possible explanation could be that HBGA B trisaccharide concentration was too low to induce the observed effects because of decreased binding affinity. Notably, NMR measurements of fully deamidated GII.4 Saga P dimers with HBGA B trisaccharide and fucose show large chemical shift perturbations around residues 370–380 [15], a region where we observe increased deuteration in presence of fucose.

We hypothesize that N373 deamidation serves as a pH and temperature dependent mechanism to control infectivity of the virus. P dimers and VLPs have been shown to be stable under low pH conditions and temperature [42], where deamidation rate is low [15]. After entering the human host via contaminated food and reaching the intestine, the rise in pH and temperature facilitates conversion of pure wildtype to partially deamidated P dimers that are still able to attach to glycans and perform the structural change potentially required for interaction with the receptor and infection of the target cell. This theory is supported by the observation of increased flexibility in the P2 cleft of iDiD GII.4 Saga P dimers compared to the wildtype [15], which is also present in GII.4 MI001 P dimers. In summary, this could imply that deamidation creates the required flexibility for host cell attachment and subsequent receptor binding. Attenuation of glycan binding in the deamidated P dimer could be counteracted by avidity due to high glycan presentation on cell surfaces *in vivo*. Native MS measurements of deamidated GII.4 Saga and MI001 P dimers also show that with increasing deamidation, dissociation into monomers occurs, whereas in NN P dimers no monomers are present (Appendix A). This could as well be linked to the increased flexibility of iDiD P dimers that weaken the dimer interface and shift the monomer-dimer equilibrium. It would be interesting to investigate whether increased flexibility is limited to the P domain or whether it is propagated into the S domain in VLPs as well, which, as a result, could destabilize the particle and prepare for uncoating.

The question remains, which advantage the evolutionary conserved N373 deamidation site provides for the most prevalent GII.4 strains over other strains. One possibility is that higher flexibility induced by deamidation indeed enables better interactions with host receptors; another possibility is that it is part of an immune escape mechanism. N373 is located in the immunodominant antibody epitope A and even minor changes in the epitope sequence during viral evolution have resulted in the loss of monoclonal antibody response [43]. From all residues in this specific epitope, N373 seems to be highly conserved over time.

On the other hand, prevalence of GII.17 strains increased over the last years [4] and based on its D377 sequence the GII.17 Kawasaki P dimer is not able to deamidate at this position. Interestingly, a N373D mutated GII.4 Saga P dimer shows similar affinities to glycans as the N373 wildtype [22] clearly illustrating that iso-aspartate is required at this position to induce the observed changes. The absence of iso-aspartate formation resulting from spontaneous deamidation could also increase stability under a wide range of pH conditions, as dissociation into monomers is less likely to occur. Increased stability under alkaline conditions has been seen for GII.17 Kawasaki VLPs; however, other strains without potential deamidation sites were less stable at alkaline pH [44,45]. GII.10 Vietnam carries a glutamine at the equivalent position 384, which, in theory, can deamidate, but deamidation is much slower and has not been observed after one year and four months of storage at 5 °C and pH 7 [15]. Nevertheless, GII.10 Vietnam clearly displays gain of structure upon glycan binding, which may cause the observed long-range effects. The larger structural dynamics could therefore be linked to cellular uptake.

Further research is required to clarify the role of N373 deamidation in the norovirus infection process. Hence, glycan binding studies with wildtype and partially deamidated VLPs will give further information about the propagation of structural changes throughout the capsid. In the future, the availability of robust cell culture systems will then allow infection studies and elucidation of the role of deamidation and glycan-binding in norovirus cell attachment *in vitro*.

## 4. Materials and Methods

### 4.1. Expression and Purification of P Dimers

GII.4 Saga 2006 (VP1 residues 225–530), GII.4 MI001 (VP1 residues 225–530), GII.10 Vietnam 026 (VP1 residues 224–538) and GII.17 Kawasaki 308 (VP1 residues 225–530) P domains (see Appendix A for VP1 sequence alignment), with GenBank accession numbers AB447457, KC631814, AF504671 and LC037415, respectively, were synthesized and purified as described elsewhere [15]. Briefly, *E. coli* BL21(DE3) were transformed with a pMal-c2x expression vector encoding the genes for ampicillin resistance, a fusion protein of maltose-binding protein, two His-tags, an HRV 3C cleavage domain and the P domain. Due to the cloning strategy, the sequences from GII.4 Saga 2006 and GII.17 Kawasaki 308 2015 P domains contain an extra GPGS sequence preceding K225, whereas GII.10 Vietnam 026 contains a GPG sequence preceding S224.

Transformed cells were grown for 3 h at 37 °C. Overexpression was induced with 1 mM isopropyl-β-d-1-thiogalactopyranoside (IPTG) at an OD_600_ value of 1.5. Incubation was continued at 16 °C for 48 h. Cells were lysed using a high-pressure homogenizer (Thermo Scientific, Waltham, MA, USA). The lysate was clarified by centrifugation and the fusion protein was purified using a Ni-NTA resin (Qiagen, Hilden, Germany). MBP and the His-tag were cleaved from the P domain using HRV 3C protease (Merck, Darmstadt, Germany). Cleaved P domain protein eluted from Ni-NTA resin and was further purified by size-exclusion chromatography using a Superdex 16/600 200 pg column (GE Healthcare, Chicago, IL, USA) in 20 mM sodium phosphate buffer (pH 7.3). Protein purity and dimer concentration were monitored by SDS-PAGE and ultraviolet absorption.

Separation of fully, partially and non-deamidated (pure N373 wildtype) GII.4 P dimer species was achieved by cation exchange chromatography using a 6 mL Resource S column (GE Healthcare) at 6 °C. After separation protein samples were prepared in 20 mM sodium acetate buffer (pH 4.9) to prevent further spontaneous deamidation and eluted using a linear salt gradient.

Wildtype P dimer samples were stored at 5 °C in the following buffers until analysis: GII.10 Vietnam: 25 mM TrisHCl, 300 mM NaCl, pH 7.3; GII.17 Kawasaki, GII.4 Saga and GII.4 MI001: 20 mM sodium acetate, 100 mM NaCl pH 4.9 (the last two pure wildtype N373). To create mixed species of wildtype (NN), partially deamidated (iDN) and fully deamidated (iDiD) GII.4 MI001 and GII.4 Saga P dimer [15], pure wildtype (NN) P dimer samples were stored at pH 7.3, which favors spontaneous deamidation, for several months. The storage conditions were 25 mM TrisHCl, 300 mM NaCl, pH 7.3, 4 °C for GII.4 MI001 P dimer and 75 mM sodium phosphate buffer, 100 mM NaCl, pH 7.3, 4 °C for GII.4 Saga P dimer.

### 4.2. Glycan Structures

Methyl α-l-fucopyranoside (α-l-Fuc-(1,O)-CH_3_) and HBGA B trisaccharide (α-d-Gal-(1,3)-[α-l-Fuc-(1,2)]-β-d-Gal-(1,O)-CH_3_) were purchased from Carbosynth, Compton, Berkshire, UK.



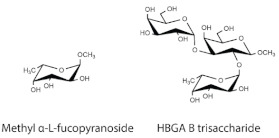



### 4.3. Native MS

Native MS measurements were performed using 3 to 4.5 μM purified P dimers. Proteins were subjected to buffer exchange to different concentrations of ammonium acetate (GII.10 Vietnam: 125 mM; GII.17 Kawasaki and GII.4 Saga: 300 mM; GII.4 MI001: 250 mM) at pH 7.5 and 4 °C via centrifugal filter units (13,000× *g*, Vivaspin 500, MWCO 10000 (Sartorius, Göttingen, Germany)) or Micro Bio-Spin 6 columns (Bio-Rad, Hercules, CA, USA) according to the manufacturers’ protocols. Mass spectra were acquired at room temperature (25 °C) in positive ion mode on an LCT mass spectrometer modified for high mass (Waters, Milford, MA, USA and MS Vision, Almere, The Netherlands) with a nano-electrospray ionization (ESI) source. Gold-coated electrospray capillaries were produced in house for direct sample infusion. Capillary and sample cone voltages were 1.20 kV to 1.35 kV and 150 to 240 V, respectively. The pusher was set to 100–150 µs. Pressures were 7 mbar in the source region and 6.2 × 10^−2^ to 6.5 × 10^−2^ mbar argon in the hexapole region. A spectrum of a 25 mg/mL cesium iodide solution from the same day was applied for calibration of raw data using the MassLynx software (Waters, Milford, MA, USA). OriginPro 2016 (Origin Lab Corporation, Northampton, MA, USA) software was used for peak integration and calculation of oligomer fractions.

### 4.4. HDX-MS

P dimers (30–50 pmol) were mixed with glycans at tenfold of the final concentration (final: 10 mM HBGA B trisaccharide, 100 mM fucose) and directly diluted 1:9 in 99% deuterated 20 mM Tris buffer (pH 7, 150 mM NaCl, 25 °C) to start the exchange reaction. After various time points (1 min, 10 min, 1 h, 8 h for most datasets, detailed information is given in the SI, HDX summary tables) the exchange reaction was quenched by 1:1 addition of ice-cold quench buffer (300 mM phosphate buffer, pH 2.3, 6 M urea), which decreased the pH to 2.3 and frozen in liquid nitrogen. As a fully deuterated (FD) control, P dimers were diluted 1:9 in 99% deuterated 20 mM Tris buffer with 150 mM NaCl and 6 M urea at pH 7, labelled for 24–72 h at room temperature and quenched as described above.

Samples were thawed, centrifuged to remove aggregates and injected onto a cooled (2 °C) HPLC System (Agilent Infinity 1260, Agilent Technologies, Santa Clara, CA, USA) equipped with a home packed pepsin column (IDEX guard column with an internal volume of 60 µL, Porozyme immobilized pepsin beads, Thermo Scientific, Waltham, MA, USA) in a column oven (25 °C), a peptide trap column (OPTI-TRAP for peptides, Optimize Technologies, Oregon City, OR, USA) and a reversed-phase analytical column (PLRP-S for Biomolecules, Agilent Technologies, Santa Clara, CA, USA). Pepsin digestion was performed online at a flow rate of 75 µL/min (0.4% formic acid in water) and peptides were trapped in the trap column. Peptides were eluted and separated on the analytical column using a 7 min gradient of 8–40% solvent B (solvent A: 0.4% formic acid in water, solvent B: 0.4% formic acid in acetonitrile) at 150 µL/min. After each run the analytical column was washed with a high percentage of solvent B. MS was performed using an Orbitrap Fusion Tribrid in positive ESI MS only mode (Orbitrap resolution 120,000, 4 microscans, Thermo Scientific, Waltham, MA, USA).

All time points were performed in three technical replicates, apart from the 8 h time point of GII.10 Vietnam with fucose, which only represents a single measurement. The triplicate measurement of GII.4 MI001 P dimer was influenced by increased peptide carryover (Figure 2), which overlaid with the lower deuterated peak distribution and led to a falsely high intensity. Therefore, a separate single-replicate measurement with additional pepsin column washing (2 M urea, 2% acetonitrile, 0.4% formic acid, pH 2.5) between sample injections was performed to minimize carryover and only deuteration differences, which are present in both datasets, are considered real.

### 4.5. Peptide and PTM Identification

Identification of peptides and post-translational modifications (PTM) was performed on non-deuterated samples using a 27 min elution gradient of 8–40% solvent B in data-dependent MS/MS acquisition mode (Orbitrap resolution 120000, 1 microscan, HCD 30 with dynamic exclusion). Precursor and fragment ions were searched and matched against a local protein database just containing the protein of interest in MaxQuant (version 1.5.7.0) using the Andromeda search engine [46]. The digestion mode was set to “unspecific” and N-terminal acetylation, deamidation, oxidation and disulfide bond formation were included as variable modifications with a maximum number of 5 modifications per peptide. Peptides between 5 and 30 amino acids length were accepted. The MaxQuant default mass tolerances for precursor (4.5 ppm) and fragment (20 ppm) ions defined for the Thermo Orbitrap instrument were used for data search. The minimum score for successful identifications was set to 20 for unmodified and 40 for modified peptides. The elution gradient was chosen in a way that wildtype and deamidated peptides could be clearly separated, as shown in a previous study [15]. For these peptides, spectra were checked manually and chromatographic peak areas where calculated in Xcalibur (Thermo Scientific, Waltham, MA, USA) to obtain a wildtype/deamidated peptide ratio.

### 4.6. HDX Data Analysis

DeutEx software (peterslab.org) was used to determine the deuterium uptake via centroid analysis [47]. Excel (Microsoft), GraphPad Prism (GraphPad Software, Inc., San Diego, CA, USA), OriginPro 2016 (Origin Lab Corporation, Northampton, MA, USA) and PyMOL (Schrödinger, New York, NY, USA) software were used for data visualization and statistical analysis. For comparison of triplicate data, a two-sided Student’s *T*-test using deuteration differences from centroid analysis was used with the α-value set to 0.05. A peptide was only considered to have a significant HDX difference if the peptide passed the *T*-test and ΔD exceeded 2x the pooled average standard deviation [48,49] of the dataset either for several time points or for the same time point in overlapping peptides. As there were some inconsistencies in the preparation of FD controls between the datasets, FD controls were not used for normalization and are not shown in the deuterium uptake plots. Only for comparison of the unbound wildtype and deamidated MI001 P dimer, the ratio of the FD controls (which were prepared under identical conditions) from both measurements was used for normalization. Additionally, a higher cut-off of ΔD > 0.42 (99% percentile calculated according to [50]) was used to account for possible day-to-day variation in the experimental conditions. Regions with significant deuterium uptake differences were mapped to existing P dimer crystal structures or the homology model (GII.4 MI001).

Deuterated spectra of peptides in certain protein regions showed bimodal peak distributions that led to lower deuteration values in centroid analysis. To validate the deuteration differences observed in centroid data analysis and to calculate relative intensities of both peak distributions, bimodal spectra of peptides representative for certain regions were analyzed by binomial fitting in HXExpress [29]. To compare relative intensities of both distributions in different states, an average over all bimodal time points for both distributions in each state was calculated for several peptides. Because of the unknown degree of carry-over in the GII.17 Kawasaki and GII.10 Vietnam experiments, relative intensities are not compared across different strains, but only between different states in the same experiment, in which the amount of carryover is considered to be unchanged. Averaged relative intensities of the first peak distribution in different peptides are presented as bar plots in Figure 3. The statistical significance of relative intensity differences of the first peak distribution in different states were analyzed with a two-sided Student’s *T*-test for each pair of states (unbound vs. ligand-bound) in an individual experiment (*p* < 0.05). Peptide coverage maps, indicating the effective peptide coverage in each HDX experiment, were plotted with MS Tools [51] and can be found in the Appendix A. Supplemental uptake plots in Appendix A represent deuteration values from automated data analysis based on the centroid of each peak distribution, which can lead to falsely low deuteration in bimodal peptides. For technical reasons, a manual analysis of individual peak distributions by binomial fitting was only possible for a representative subset of peptides from each region. Corrected uptake plots for peptides from regions with significant deuteration differences (Table 1) generated by binomial fitting are shown in Appendix A and Woods’ plots for all datasets are shown in Appendix A.

### 4.7. Experimental Design and Statistical Rationale 

The rationale for experimental design and data analysis is based on HDX-MS community-recommendations [52]. In brief, sample quality was assessed with native MS and HDX-MS conditions were optimized for maximum sequence coverage and detection sensitivity. Labeling time points were chosen to cover 3–4 orders of magnitude. Three independent labeling reactions were performed for each time point and the level of back exchange was assessed with a fully deuterated protein control as well as a mix of deuterated model peptides. Details about the peptide identification method, statistical analysis with Student’s *T*-test and color mapping procedure are given in the individual methods sections. Fragmentation spectra for identification of deamidated peptides are given in the Appendix A. All HDX-MS data were manually inspected and exchange differences in bimodal peak distributions have been validated by binomial fitting. HDX summary tables with detailed information about experimental conditions and statistics, as well as deuterium uptake plots for each dataset, can be found in the Appendix A.

### 4.8. Structure and Sequence Alignment

GII.4 Saga, GII.4 MI001, GII.10 Vietnam and GII.17 Kawasaki VP1 protein sequences were aligned with T-Coffee [53] and visualized with Jalview (version 2.11.0) [54]. A phylogenetic tree was created in Jalview with BLOSUM62 and Neighbor joining. GII.4 Saga (pdb 4X06), GII.10 Vietnam (pdb 3ONY) and GII.17 Kawasaki (pdb 5F4O) P dimer crystal structures were superimposed in PyMOL.

### 4.9. Homology Modeling of GII.4 MI001 P Dimer Structure

The SWISS-MODEL template library [55,56] SMTL version 2019-10-24, PDB release 2019-10-18 was searched with BLAST [57] and HHBlits [58] for evolutionary related structures matching the GII.4 MI001 P dimer target sequence. Based on the search results, the GII.4 Farmington Hills P dimer structure (pdb 4OOV, 94% sequence identity) was used for model building. Models were built based on the target-template alignment using ProMod3. The global and per-residue model quality has been assessed using the QMEAN scoring function [59]. The resulting GMQE (Global Model Quality Estimation) was 0.99 and QMEAN was 0.57 indicating very good accuracy and quality of the model structure (Appendix A).

### 4.10. MD Simulations

We performed molecular dynamics (MD) simulations on the following proteins: GII.4 Saga (pdb 4X06), GII.4 Saga containing a deamidated P domain (pdb 6H9V) (iDN), GII.4 MI001 (pdb 4OOV), GII.10 Vietnam (3ONU), GII.17 Kawasaki (5F4O). An additional proteoform was generated comprising fully deamidated P dimers at residue 373 in both peptide chains, based on GII.4 Saga (pdb 6H9V) (iDiD). All pdb-structures were refined by adding missing atoms and residues using the UCSF Chimera tool (version 1.14) [60]. GII.4 Saga P dimers were additionally simulated with α-l-methyl-fucose (F) ligands to explore a potential influence of deamidation on protein dynamics. Hence, the amount of systems is expanded to include wildtype GII.4 Saga P dimers (pdb 4X7C) (NN) with one (N_F_N) and two (N_F_N_F_) fucose ligands, iDN P dimers with one fucose complexing each individual chain (iD_F_N and iDN_F_) and two fucoses (iD_F_N_F_) and further include iDiD P dimers with one (iD_F_iD) and two (iD_F_iD_F_) fucose ligands.

All MD simulations were performed using Gromacs on the Rackham cluster of the Uppsala Multidisciplinary Center for Advanced Computational Science (UPPMAX) and the Kebnekaise cluster at the High Performance Computing Center North (HPC2N) [61]. The amber99sb force field was utilized for all simulations [62], modified to include parameters for iD and F [63]. The MkVsites tool provided virtual sites and dummy-mass constructions for F [64]. Structures were placed in a dodecahedral box under periodic boundary conditions, solvated using the TIP3P water model [65] and neutralized in a 154 mM saline solution by adding NaCl. Protonation states of all systems were based on the sidechains’ pKa at pH 7. Each system was minimized using the steepest descent algorithm, followed by a 100 ps simulation with applied position-restrains. Temperature and pressure were maintained at 300 K and 1 bar by the v-rescale thermostat and the Parrinello–Rahman barostat, with coupling constants of 50 fs for both [66,67,68]. Neighbor lists were updated every 10 steps. The particle mesh Ewald algorithm was used for Coulomb interactions, with a real-space cut-off of 1.0 nm [69,70]. The systems were allowed to relax for 100 ns with a 5 fs time step, extracting one frame every 10th ns as starting structures for the production runs. Final simulations were performed for ten 100 ns production runs at a 5 fs time step. As such, each of the 14 systems was simulated in 10 replicates from different starting structures, resulting in an aggregated simulation time of 1 µs per system, making 14 µs in total for all systems.

The root-mean-square deviation (RMSD) and fluctuation (RMSF), as well as the solvent accessible surface area (A_sas_), were calculated to analyze the behavior of each system. The RMSD was computed with the first frame of the individual trajectory as reference structure. The trajectories of the ten replicas were combined to a single trajectory, of which the average structure was calculated with the Gromacs software package. This average structure was then taken as reference for RMSF calculations, as this most accurately represents the standard deviation of the individual atomic positions. To further support the RMSF calculations, we computed the A_sas_ of the protein backbone for the initial conformation (pdb-structure) and the final production simulations, of which latter was combined to an average representation of the area over all ten production replicas. The resulting values were subtracted from each other, to eventually visualize an increase or decrease of the A_sas_ after 100 ns. The RMSF and A_sas_ values for the different P dimer strains were aligned with the sequence to compare each residue between the dimers.

## Figures and Tables

**Figure 1 molecules-26-02125-f001:**
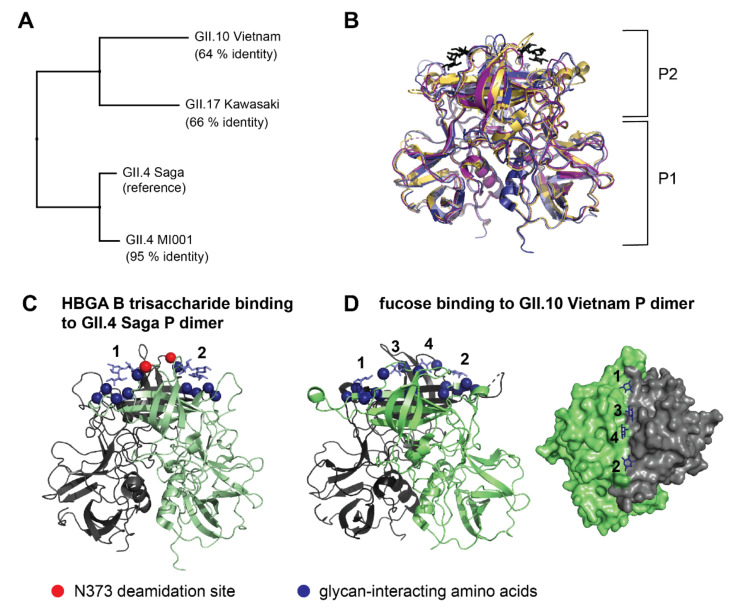
Comparison of human norovirus VP1 sequences (**A**) and P dimer structures with HBGA B trisaccharide (black) binding to the canonical binding site (**B**). P dimer sequences of three virus strains were aligned to the already investigated [15] GII.4 Saga strain as reference. For the GII.4 MI001 P dimer, a homology model was created for comparison using the GII.4 Farmington Hills P dimer structure with 94% sequence identity to MI001 as reference. Crystal structures of GII.4 Saga (pdb 4X06, dark blue), GII.10 Vietnam (pdb 3ONY, yellow), GII.17 Kawasaki (pdb 5F4O, purple) and the homology modelled structure of GII.4 MI001 (light blue) were superimposed in PyMOL with the following RMSDs to GII.4 Saga: 7.4 Å (GII.10 Vietnam), 6.5 Å (GII.17 Kawasaki), 1.6 Å (GII.4 MI001). P1 and P2 indicate the respective domains of the P dimer. (**C**) HBGA B trisaccharide binding to the two canonical binding sites of the GII.4 Saga P dimer (pdb 4x06). Amino acids contributing to the binding site (blue spheres) and the GII.4 N373 deamidation site (red sphere) are indicated in both monomers of the P dimer (green, grey). (**D**) Fucose binding to the GII.10 Vietnam P dimer (canonical binding sites: 1,2; additional binding sites: 3,4). The surface representation shows the glycan binding cleft in top-view (pdb 4z4s).

**Figure 2 molecules-26-02125-f002:**
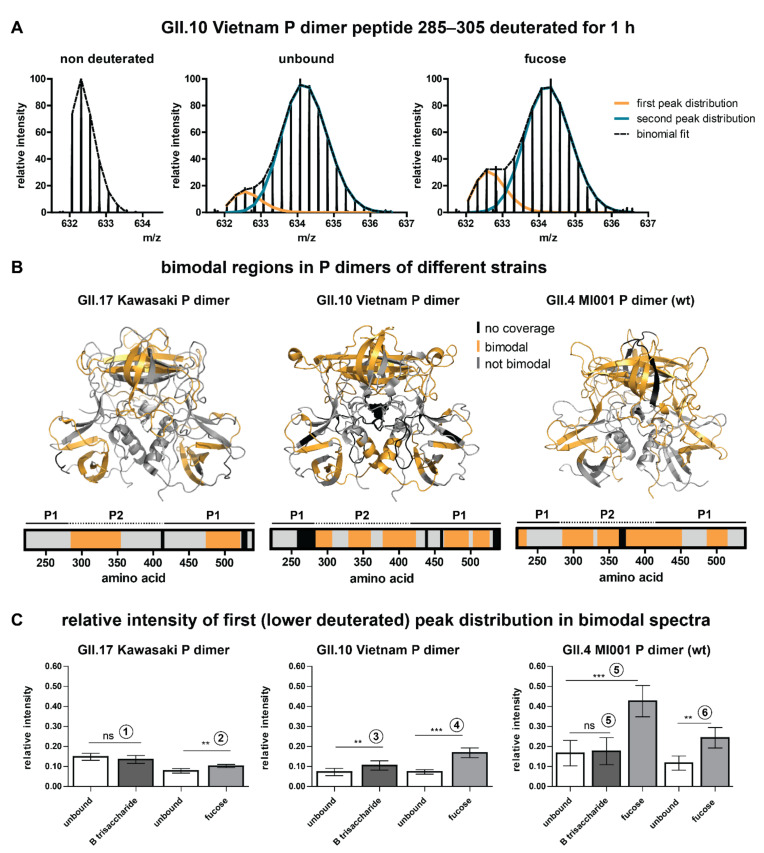
Bimodal peak distributions in some peptides point towards the existence of a second, more compact subpopulation. (**A**) Bimodality in deuterated spectra of an exemplary peptide of the GII.10 Vietnam P dimer. Bimodality occurs in both the unbound and the glycan bound state. Single peak distributions can be separated by binomial fitting. (**B**) Regions in P dimers of different strains that show bimodality in deuterated spectra (orange). Bimodality mainly occurs in the P2 domain and the lower part of the P1 domain in all strains. The amino acid numbering is based on the VP1 sequence. (**C**) Relative intensity of the first (lower deuterated) peak distribution for different strains and individual experiments (1–6) calculated by binomial fitting. Relative intensity of the first peak distributions stays constant over time, so the averages over all bimodal time points for all states were calculated for several peptides of different protein regions and combined into bar graphs. The error bar represents the standard deviation of the average relative intensity calculated from N ≥ 5 peptides. Significant differences between the unbound and the glycan bound state were assessed using a two-sided Student’s *T*-test for each pair in an individual experiment. P values are indicated by asterisks: *p* < 0.001 (***), *p* < 0.01 (**) and not significant (ns). Note that in experiments 1–5 different degrees of peptide carryover from the pepsin column led to a variation of the relative intensity of the lower deuterated peak distribution between the experiments, but not within a certain experiment. In experiment 6 an additional wash step eliminating carryover was applied, so that the lower deuterated peak distribution mainly reflects the more compact protein subpopulation.

**Figure 3 molecules-26-02125-f003:**
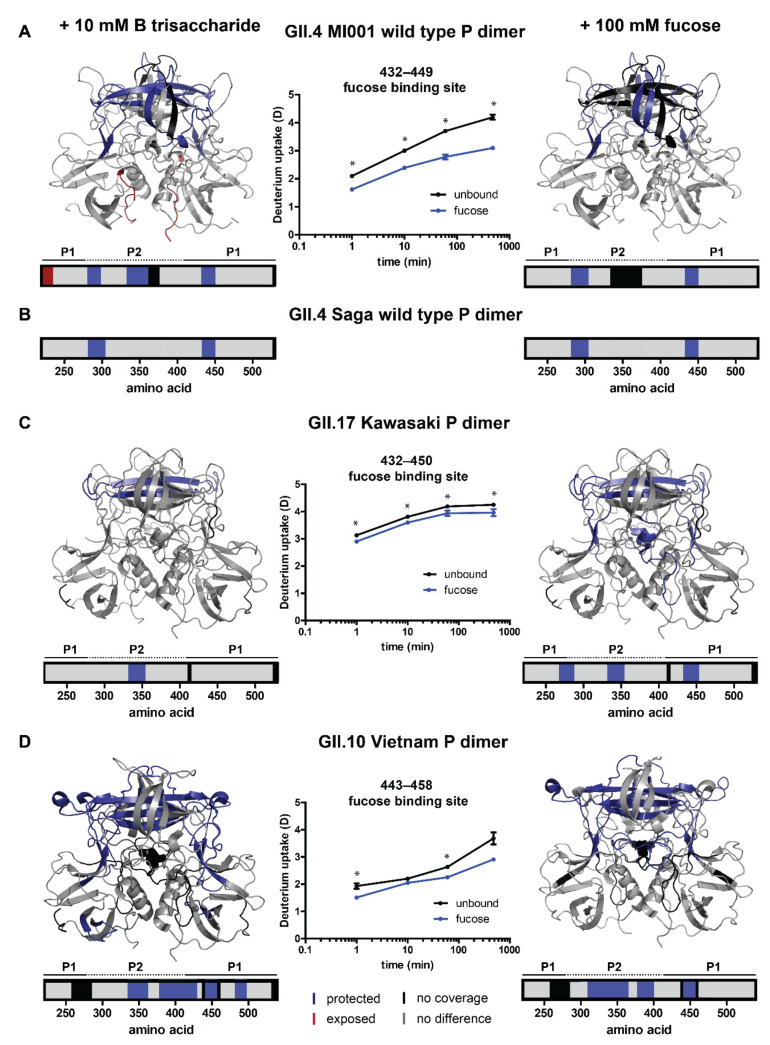
HDX differences upon glycan binding in human GII.4 MI001 (**A**), GII.17 Kawasaki 308 (**C**) and GII.10 Vietnam (**D**) norovirus P dimers. (**B**) Protected regions in wildtype GII.4 Saga P dimers are shown for comparison [15]. Depicted are protein regions with significant deuterium uptake differences in the second (main) peak distribution between unbound P dimers and P dimers with either 10 mM HBGA B trisaccharide or 100 mM fucose (*p* < 0.05, Student’s *T*-test and ΔD > 2x pooled average SD, Appendix A). Deuterium uptake plots show significant (*) protection of the canonical fucose binding site. The deuteration difference at the 8 h time point of GII.10 Vietnam is not considered significant based on the applied criteria because the fucose state only represents a single measurement and thus no Student’s *T*-test can be applied. In case of bimodal spectra, deuteration differences were manually validated by binomial fitting. Error bars indicate the standard deviation. Bar graphs and colored structures illustrate regions of P dimers, which get significantly more protected (dark blue) or exposed (red) upon interaction with glycans. Areas colored in grey showed no significant difference in the chosen HDX time regime and black areas have no peptide coverage. P1/P2 refers to the two domains of the P dimer (shown in Figure 1).

**Figure 4 molecules-26-02125-f004:**
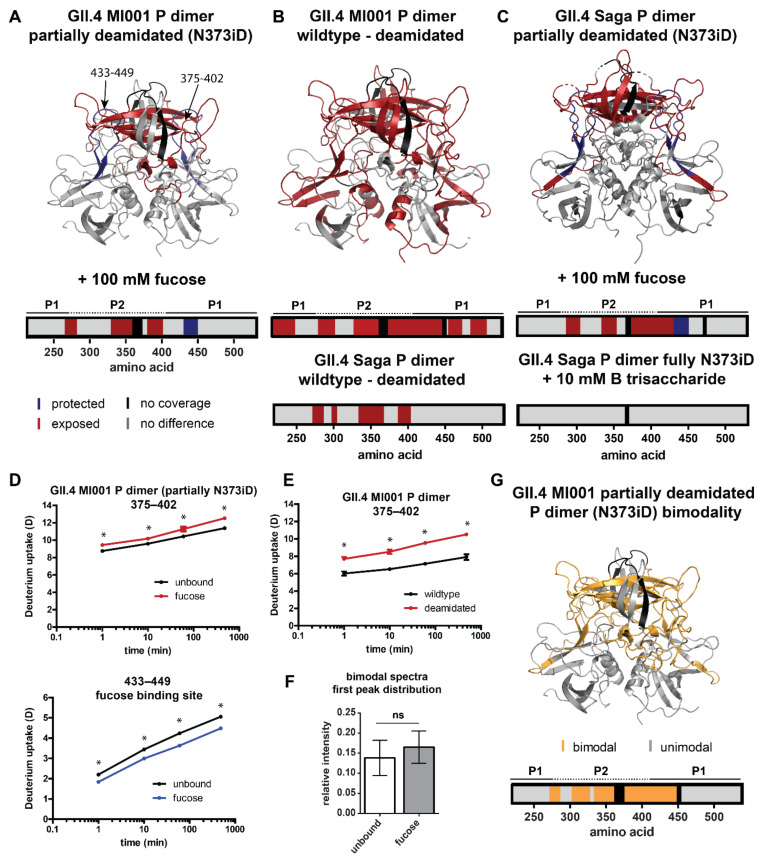
Significant HDX differences in partially deamidated (N373iD) GII.4 P dimers. A/C/D) Fucose can still bind to the canonical glycan binding site in partially deamidated GII.4 MI001 (**A**) and GII.4 Saga (**C**) P dimers (pdb 6H9V). In contrast to the wildtype N373 P dimer, parts of the P2 domain get more exposed upon interaction with 100 mM fucose. HDX differences in fully deamidated GII.4 Saga P dimers in presence of 10 mM HBGA B trisaccharide are shown for comparison [15]. (**B**,**E**) The partially deamidated GII.4 MI001 N373iD P dimer shows a higher deuterium uptake in large parts of the structure, which points towards higher flexibility, like in the fully deamidated GII.4 Saga P dimer shown for comparison [15]. (**D**,**E**) Deuterium uptake plots exemplarily highlight statistically significant (*) deuteration differences in partially deamidated GII.4 MI001 P dimers in presence of fucose (**D**) and in comparison to the wildtype protein (**E**). All uptake plots show deuteration values of the second (main) peak distribution in case of bimodality. (**F**,**G**) Bimodality occurs in similar regions as for the wildtype P dimer, implying that the more protected subpopulation is also present in the partially deamidated sample (**G**). In contrast to the native P dimer, the relative intensity of the first peak distribution does not significantly (ns) increase under fucose treatment as shown in the bar plot in panel (**F**) (for statistics refer to description of Figure 3). Error bars represent the standard deviation.

**Figure 5 molecules-26-02125-f005:**
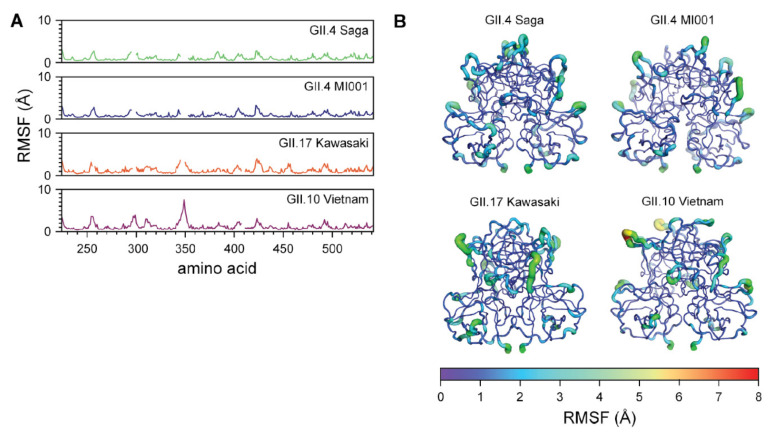
Difference in fluctuations between the P dimer strains propose increased flexibility in the protein chains in absence of a ligand. (**A**) RMSF data of GII.4 Saga, GII.4 MI001, GII.17 Kawasaki and GII.10 Vietnam, simulated for a total of 1 µs each, reveal different protein chain dynamics between the strains, with most prominent peak around residue 350. Gaps in the data originate from alignment of the norovirus P dimer sequences. (**B**) RMSF values of GII.4 Saga, GII.4 MI001, GII.17 Kawasaki and GII.10 Vietnam visualized in the structures, highlighting residues with increased fluctuations during the simulations in absence of ligand.

**Figure 6 molecules-26-02125-f006:**
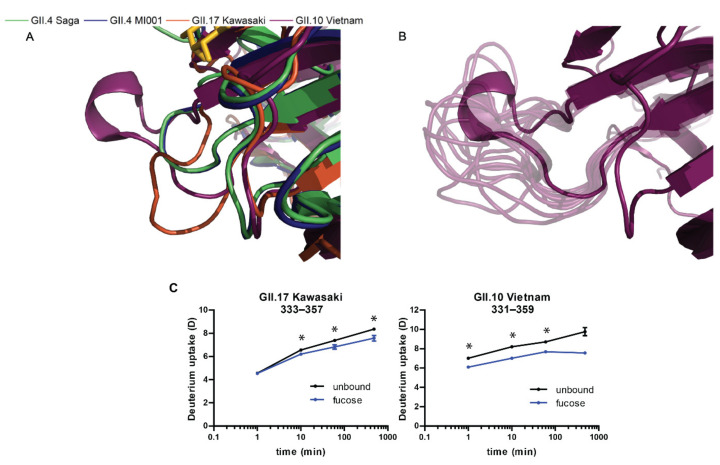
Structural rearrangement of the glycan binding site in absence of fucose. (**A**) Zoomed in image of the overlaid P domain structures around residue 350, complexed with fucose. The presence of fucose stabilizes a small helix, forming an ordered pocket in the GII.10 Vietnam strain (purple). (**B**) Absence of fucose results in a high flexibility of the GII.10 Vietnam pocket and a loss of the short helix, as shown in the snapshots of different conformations throughout the simulation (light purple). These snapshots were taken from every 10th frame of a single trajectory. (**C**) Deuterium uptake plots show significant protection (*) around residue 350 in presence of fucose, with largest deuteration differences in GII.10 Vietnam. Error bars represent the standard deviation.

**Table 1 molecules-26-02125-t001:** Comparison of protected residues in HDX with known glycan interactions in crystal structures. The canonical binding sites 1/2 are conserved for many strains and glycans, binding sites 3/4 have so far only been detected in GII.10 Vietnam. For GII.17 Kawasaki only crystal structures with fucose and A trisaccharide are available, so protected residues for B trisaccharide are compared to binding sites seen for A trisaccharide. P dimer chain annotations are given for fucose binding sites 1 and 3. No crystal structure is available for GII.4 MI001 P dimers, so binding sites are marked as unknown (NA). Protected residues for wildtype GII.4 Saga P dimers [15] are shown for comparison.

P Dimer Dataset	Protected Residues in HDX	Fucose Binding Site 1/2	Fucose Binding Site 3/4
GII.10 Vietnam + 100 mM fucose [14]	311–336	-	-
	337–364	N355 (chain A) R356 (chain A)	E359 (chain A)
	379–399	D385 (chain A)	W381 (chain A)
	442–458	G451 (chain B) Y452 (chain B)	L449 (chain A)
GII.10 Vietnam + 10 mM HBGA B trisaccharide [12]	336–361	N355 (chain A) R356 (chain A)	-
	379–428	D385 (chain A)	-
	440–458	G451 (chain B) Y452 (chain B)	-
	483–496	-	-
GII.17 Kawasaki + 100 mM fucose [13,27]	269–286	-	-
	333–353	T348 (chain A) R349 (chain A)	-
	434–452	G443 (chain B) Y444 (chain B)	-
	-	D378 (chain A)	-
GII.17 Kawasaki + 10 mM HBGA B trisaccharide [13]	333–353	T348 (chain A) R349 (chain A)	-
	-	G443 (chain B) Y444 (chain B)	-
GII.4 MI001 + 100 mM fucose	283–303	NA	NA
	434–449	NA	NA
GII.4 MI001 + 10 mM HBGA B trisaccharide	283–298	NA	NA
	333–353	NA	NA
	434–450	NA	NA
GII.4 Saga + 100 mM fucose / GII.4 Saga + 10 mM HBGA B trisaccharide [15,21]	283–303	-	-
	no coverage	D374 (chain A)	-
	434–449	G443 (chain B) Y444 (chain B)	-


## Data Availability

Full HDX data tables as well as MS raw data and peptide identification results have been deposited to the ProteomeXchange Consortium [71] via the PRIDE [72] partner repository (dataset identifier PXD019884). Annotated fragment ion spectra of all protein/peptide identifications can be viewed with MS-Viewer using the respective search keys given in the Appendix A.

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
