# Peer review of "Glycan-Induced Protein Dynamics in Human Norovirus P Dimers Depend on Virus Strain and Deamidation Status"

_molecules, 2021, doi:10.3390/molecules26082125_

Round 1

Reviewer 1 Report

This manuscript describes the use of HDX-MS to determine glycan binding differences between strains of the norovirus as well as understanding the effect of deamidation on protein dynamics. The data shows a bimodal distribution for several peptides indicating the presence of a compact subpopulation in the sample, however, this compact structure is not detected by native MS or size exclusion chromatography presumably due to its low abundance. Some differences in regions of protection are observed in the different strains although all have protection in the canonical glycan binding site at G443 and Y444). In addition, studies of the deamidated protein show differences in protein flexibility upon deamidation suggesting a role for this modification in the virus pathology.

There are several places within the manuscript that are confusing and require clarification prior to publication.

-Page 8 lines 250-253 the authors discuss the canonical glycan binding site (G443, Y444) and then discuss canonical binding site (G451, Y452) on line 253. It is not clear what the difference is between these two canonical binding sites. I understand there are two fucose binding sites but there should be some description of each one that does not include calling both of them canonical binding site.

-Also, on line 253 of page 8, authors should list the residues for the B-sheet region in the binding cleft so the reader can easily correlate this back to Figure 3.

-Figure 4D has too many panels that are not well coordinated or discussed together in the results. This should be broken up into individual letters. For example, page 8 lines 270-273 discuss main peak distribution. I actually don’t even know what this is referring to in Figure 4D. What is meant by main peak distribution? If you are discussing this for particular peptides highlighted in this figure that needs to be stated not using this general term.

-The authors hypothesize that the increased flexibility in the deamidated proteins distant from the glycan binding site could weaken the dimer interface leading to monomer formation. Authors should provide some data on how far away these regions are from the dimer interface.

-Figure 5C should be with Figure 6 rather than Figure 5. That is where it is discussed in the text and is confusing in its current location.

Author Response

This manuscript describes the use of HDX-MS to determine glycan binding differences between strains of the norovirus as well as understanding the effect of deamidation on protein dynamics. The data shows a bimodal distribution for several peptides indicating the presence of a compact subpopulation in the sample, however, this compact structure is not detected by native MS or size exclusion chromatography presumably due to its low abundance. Some differences in regions of protection are observed in the different strains although all have protection in the canonical glycan binding site at G443 and Y444). In addition, studies of the deamidated protein show differences in protein flexibility upon deamidation suggesting a role for this modification in the virus pathology.

There are several places within the manuscript that are confusing and require clarification prior to publication.

-Page 8 lines 250-253 the authors discuss the canonical glycan binding site (G443, Y444) and then discuss canonical binding site (G451, Y452) on line 253. It is not clear what the difference is between these two canonical binding sites. I understand there are two fucose binding sites but there should be some description of each one that does not include calling both of them canonical binding site.

We thank the reviewer for the positive and constructive feedback. There is indeed only one canonical binding site per monomer, which is now highlighted in Fig.1. The exact amino acid position differs between strains with a shift of 8 amino acids due to the loop insertion in the GII.10 Vietnam. We have clarified in the text, which amino acid positions belong to which strain. In addition, we have moved Table 1 into the results section so that regions protected in HDX can be directly compared with the know glycan binding sites.

-Also, on line 253 of page 8, authors should list the residues for the B-sheet region in the binding cleft so the reader can easily correlate this back to Figure 3.

This is a helpful suggestion and we now reference the residues throughout the text.

-Figure 4D has too many panels that are not well coordinated or discussed together in the results. This should be broken up into individual letters. For example, page 8 lines 270-273 discuss main peak distribution. I actually don’t even know what this is referring to in Figure 4D. What is meant by main peak distribution? If you are discussing this for particular peptides highlighted in this figure that needs to be stated not using this general term.

We have restructured Fig.4 to avoid the confusing nature of panel 4D. In the text and figure legend, it is now also clarified that it refers to the second highly exchanging peak distribution observed for bimodal peptides.

-The authors hypothesize that the increased flexibility in the deamidated proteins distant from the glycan binding site could weaken the dimer interface leading to monomer formation. Authors should provide some data on how far away these regions are from the dimer interface.

Fig.1 now includes additional panels highlighting the two monomers, canonical binding site and deamidating Asn. The P2 domain forms a major part of the dimer interface, which underscores our hypothesis that increased dynamics in the P2 domain of deamidated P domains increase dimer dissociation, which has been clarified in the results.

-Figure 5C should be with Figure 6 rather than Figure 5. That is where it is discussed in the text and is confusing in its current location.

We moved panel 5C to Fig. 6 to align with the discussion.

Reviewer 2 Report

The manuscript by Dulfer et al deals with a characterization of glycan binding properties of norovirus VP1 protein (its P domain) and the influence of protein deamidation on these properties using mass spectrometric techniques. Such investigation is surely of importance and may have further functional implications in other studies aimed at e.g. virus binding inhibition or at difference in viral entry, infectivity, etc. The paper uses well established and to a large part properly technically employed technologies. I find it suitable for the Molecules journal and consider it as a basis for future biologically more-relevant studies. Therefore, I think it might be published upon some minor revisions that are based on improved data presentation and text editing.

Even though the results are solid and point to important molecular events I am finding their presentation overcomplicated. I think that this is the major point that must be addressed before the paper is accepted for publication.

Pls provide structure of the protein where the two domains (p1 and P2) will be differentially colored and all important features will be shown – e.g. glycan binding site(s), deamidation sites, P2 binding cleft and other potential features that are somehow highlighted in the text.

The study contains a large portion of data (provided in the Supplementary material) which are not presented in the best form and that makes them difficult to follow / compare and draw the conclusions. I think that the current supplement should be separated into two files. One containing “raw data” such as all uptake plots and one covering true supplement – hence figures supporting the main figures or extending them. Here HDX data should be presented in a form that allow general overview of the profiles or differences. If all redundant data are included then Wood’s plot or butterfly plot may be the most appropriate form. If the data are recalculated to the narrowest possible region then heat maps or differential heat maps would do the trick. 

I acknowledge the fact that authors correctly address the issue of back-exchange and identified some degree of it. Even though it is not fully correct to use such data, instead of full dataset reanalysis with proper column washing, their claim that in comparison experiment the contribution of carry-over can be partially disregarded is not completely incorrect. Also, if they build their conclusions on data with differences that are far enough from the significance level, the threat of carry-over effects is not a significant issue.
However, I do not think that the section related to bimodality deserves to be so widely elaborated. In my point of view it should be shortened and suggest also reduction of the related part of the discussion. What is the role and significance of P particles?

Were the deamidated peptides separated chromatographically enough so that the overlap in the MSMS spectrum can be completely ruled-out? Are the details of the MS peaks devoid of any contribution of deamidated species? This is important not only for identification but also for the subsequent HDX. It should be somehow addressed in the text. Is it possible to see uptake plots for unmodified (N-containing) and deamidated version of peptides covering deamidation sites?

It is quite surprising that the deamidated sites (N at positions  239, 373, 448) was possible while  these residues/regions are not covered in the maps. E.g. GII.4 Saga N373 – not covered in FigS16. I think this needs some explanation.

The HDX coverage maps show a reasonable degree of redundancy but it is not clear how this redundancy in the data was used. E.g. was it employed in recalculation of the data into shorter segments based on the overlapping peptides? If not, what was the decision in selecting some peptides/regions - e.g. Table 1 GII.10 Vietnam 442-458 (this part is covered by 5 peptides, all having common end – 458). I think the overall representation should reflect the benefit of redundancy and use either recalculation to shorter segments or visualization that allows display of all the acquired data points.

What is the reason for very high temperature used during digestion step? I assume that it is also reason for unusually high back-exchange. Up to 64%!  This may mask some differences due to high deuterium loss. However if the back-exchange was calculated on the basis of FD sample, then it I found it more likely that the FD sample was not simply deuterated enough. Common mistake in FD preparation under denaturing conditions is inclusion of non-deuetrated urea at high concentration which results in much lower D%. Are the authors sure about their statement on lines 664-665 that the buffer was 99% deuterated? 

I think that if the overall deuteration levels for same or sequence similar peptides are more or less same across all datasets, then the FD control is not needed and should be omitted as its current form (seemingly irreproducible) makes the data and their comparison complicated and comparison experiments do not require FD.

Is there any explanation why the region 259-284 is not covered in GII.10 Vietnam while it is nicely covered in other P variants? Is that due to insufficient reduction of the disulfide bonds? Does it make sense to run the search of the MSMS data using Cys (-H) modification – my help if the SS bonds within this area are “intrapeptide”. Interestingly, also other Cys in this map are not covered which may point to insufficient reduction.

Conversion of Asn into isoAsp is described detrimental for glycan binding (lines 517-519, ref22). What is the state of deamidated Asn in this study? Any proof if all deamidated Asn are present as isoAsp?

Would it be possible to label sites of deamidation in the supplementary maps? If possible, indicate also glycan binding sites (even if they are putative or identified within this paper).
Overall, some highlight of these sites would be beneficial also in the other figures – 2, 3, 4, 5, S1, etc. Simply wherever applicable since the modification is of high importance for the message of this paper.
I also suggest use of colors in the supplement. I assume that the MStools that were used to generate these figures allows that. Are those peptides used in HDX or all peptides that can be identified (likely not, given my comment above related to coverage of the sites with deamidated residues)?

Is there any publication related to the DeutEx software?

Fig S16 – I think that the numbering here is shifted by 1. All proteins are equally numbered (based on Fig S1 alignment) up to position 298. Hence the numbering should also start with 220 like in the other maps. However this numbering is likely incorrect as it incorporates four initial amino acids from the expression vector (GPGS). In my point of view, the numbering should start on the fifth amino acid (Lys) in all maps.

Does it make sense to show different conditions for peptide coverage in FigS12-16? Please explain! If the comparison is targeted the just peptides found in all relevant conditions are more appropriate dataset as the aim is anyway comparison and that needs same peptides in both (or all) conditions under comparison.

What does it mean dimer wt vs. deam in Fig S15? Is that a comparison of two states or their difference?

Author Response

The manuscript by Dulfer et al deals with a characterization of glycan binding properties of norovirus VP1 protein (its P domain) and the influence of protein deamidation on these properties using mass spectrometric techniques. Such investigation is surely of importance and may have further functional implications in other studies aimed at e.g. virus binding inhibition or at difference in viral entry, infectivity, etc. The paper uses well established and to a large part properly technically employed technologies. I find it suitable for the Molecules journal and consider it as a basis for future biologically more-relevant studies. Therefore, I think it might be published upon some minor revisions that are based on improved data presentation and text editing.

Even though the results are solid and point to important molecular events I am finding their presentation overcomplicated. I think that this is the major point that must be addressed before the paper is accepted for publication.

Pls provide structure of the protein where the two domains (p1 and P2) will be differentially colored and all important features will be shown – e.g. glycan binding site(s), deamidation sites, P2 binding cleft and other potential features that are somehow highlighted in the text.

The reviewer is thanked for constructive feedback, which further shaped the manuscript. Fig.1 now includes additional panels highlighting the two monomers, canonical binding site, binding cleft and deamidating Asn. P1 and P2 domain are already depicted in Fig. 1B.

The study contains a large portion of data (provided in the Supplementary material) which are not presented in the best form and that makes them difficult to follow / compare and draw the conclusions. I think that the current supplement should be separated into two files. One containing “raw data” such as all uptake plots and one covering true supplement – hence figures supporting the main figures or extending them. Here HDX data should be presented in a form that allow general overview of the profiles or differences. If all redundant data are included then Wood’s plot or butterfly plot may be the most appropriate form. If the data are recalculated to the narrowest possible region then heat maps or differential heat maps would do the trick.

We agree with the reviewer that the supplement can be arranged better. However, we also believe that it is best to have all supplementary data in a single file. We have introduced two subsections for the “true” and raw data supplement. Content is linked so that readers can navigate through the pdf.

In the true supplement comprising the front part, we have rearranged a few items to support conclusions and also included a Woods’ plot on the centroid data, in which the bimodal regions, regions with significant deuteration differences and known glycan binding sites are further highlighted, following the table showing regions with changes and corresponding uptake plots.

I acknowledge the fact that authors correctly address the issue of back-exchange and identified some degree of it. Even though it is not fully correct to use such data, instead of full dataset reanalysis with proper column washing, their claim that in comparison experiment the contribution of carry-over can be partially disregarded is not completely incorrect. Also, if they build their conclusions on data with differences that are far enough from the significance level, the threat of carry-over effects is not a significant issue.
However, I do not think that the section related to bimodality deserves to be so widely elaborated. In my point of view it should be shortened and suggest also reduction of the related part of the discussion. What is the role and significance of P particles?

We fully agree and have shortened the results description and discussion on bimodality. This hopefully provides a more focused view on how it influences data analysis. P particles are likely not of biological significance due to the lack of free P domains during infection but have been discussed as e.g. vaccine candidates. Nevertheless, they are a logic explanation for a defined low exchanging subpopulation that would be common across the majority of strains studied. We hope that this becomes clearer in the streamlined discussion.

Were the deamidated peptides separated chromatographically enough so that the overlap in the MSMS spectrum can be completely ruled-out? Are the details of the MS peaks devoid of any contribution of deamidated species? This is important not only for identification but also for the subsequent HDX. It should be somehow addressed in the text. Is it possible to see uptake plots for unmodified (N-containing) and deamidated version of peptides covering deamidation sites?

Thanks for the question, yes, the wildtype and deamidated peptides are separated significantly due to the altered hydrophobicity. We included a short notion in the methods (line 685) that this can be seen in the Nat Commun manuscript from 2019. The peptide affected by deamidation in GII.4 noroviruses can be detected and identified as is also shown in the supplement from the fragmentation spectra. However, intensity split-up in deamidated samples resulted in exclusion of the respective peptides in HDX analysis (lines 130-132).

It is quite surprising that the deamidated sites (N at positions  239, 373, 448) was possible while  these residues/regions are not covered in the maps. E.g. GII.4 Saga N373 – not covered in FigS16. I think this needs some explanation.

As outlined above, the deamidated peptides were covered during peptide mapping, where we had almost complete coverage, but signal was insufficient for HDX. This is now clarified in methods and also in the legend of the coverage maps.

The HDX coverage maps show a reasonable degree of redundancy but it is not clear how this redundancy in the data was used. E.g. was it employed in recalculation of the data into shorter segments based on the overlapping peptides? If not, what was the decision in selecting some peptides/regions - e.g. Table 1 GII.10 Vietnam 442-458 (this part is covered by 5 peptides, all having common end – 458). I think the overall representation should reflect the benefit of redundancy and use either recalculation to shorter segments or visualization that allows display of all the acquired data points.

We agree with the reviewer that using the redundancy would be beneficial and in general overlapping peptides do show consistent behaviour. Due to many areas being affected by bimodality, we refrained from recalculation as this would only be possible for the centroid data. Manual reanalysis is laborious, so we have limited it to the most significant peptides covering a certain area and depicted the plots in Tab. S 3.

What is the reason for very high temperature used during digestion step? I assume that it is also reason for unusually high back-exchange. Up to 64%!  This may mask some differences due to high deuterium loss. However if the back-exchange was calculated on the basis of FD sample, then it I found it more likely that the FD sample was not simply deuterated enough. Common mistake in FD preparation under denaturing conditions is inclusion of non-deuetrated urea at high concentration which results in much lower D%. Are the authors sure about their statement on lines 664-665 that the buffer was 99% deuterated? 

I think that if the overall deuteration levels for same or sequence similar peptides are more or less same across all datasets, then the FD control is not needed and should be omitted as its current form (seemingly irreproducible) makes the data and their comparison complicated and comparison experiments do not require FD.

The temperature during digestion was 25°C, which is not uncommon. In most datasets, the back exchange was just above 30%. The dataset with the high back exchange has indeed likely an issue with a faulty prepared FD control resulting in an arbitrarily high back exchange.

Moreover, as the reviewer correctly notes, deuteration levels were consistent across datasets. We therefore omitted the FD control as suggested by the reviewer and added a notion in the supplement legends for the raw data on back exchange.

Is there any explanation why the region 259-284 is not covered in GII.10 Vietnam while it is nicely covered in other P variants? Is that due to insufficient reduction of the disulfide bonds? Does it make sense to run the search of the MSMS data using Cys (-H) modification – my help if the SS bonds within this area are “intrapeptide”. Interestingly, also other Cys in this map are not covered which may point to insufficient reduction.

Thanks for the suggestion, P dimers are not known to form disulphide bonds and Cysteine modification was included in the MS data search. As for some other regions, peptides covering this region were identified during peptide mapping, but had to be excluded in the HDX analysis due to low data quality. This information has been added to the peptide map legend.

Conversion of Asn into isoAsp is described detrimental for glycan binding (lines 517-519, ref22). What is the state of deamidated Asn in this study? Any proof if all deamidated Asn are present as isoAsp?

We can only speculate whether Asn is deamidated into Asp or isoAsp as the distinction is not readily made from MS data. The distinction can be made in NMR as was done for the GII.4 Saga. The specific conversion into isoAsp in that case probably depends on the structural arrangement in the loop. This is highly conserved across GII.4 and also the GII.4 MI001 homology model would suggest a similar structure. Furthermore, both strains show nearly identical deamidation kinetics (Creutznacher, R., et al. Viruses, 2021. 13(3): p. 416) and behaviour in our HDX-MS. Thus, while isoAsp is likely, we have no prove and therefore do not make a statement on this in the manuscript.

Would it be possible to label sites of deamidation in the supplementary maps? If possible, indicate also glycan binding sites (even if they are putative or identified within this paper).
Overall, some highlight of these sites would be beneficial also in the other figures – 2, 3, 4, 5, S1, etc. Simply wherever applicable since the modification is of high importance for the message of this paper.

We have adapted Fig.1 to highlight sites of interest. Reference is now always given to residue numbers in the text when we refer to a certain region. Deamidated peptides (see above) are not covered in HDX analysis but we have indicated the canonical binding sites and deamidation sites in the Woods plots and coverage maps.

I also suggest use of colors in the supplement. I assume that the MStools that were used to generate these figures allows that. Are those peptides used in HDX or all peptides that can be identified (likely not, given my comment above related to coverage of the sites with deamidated residues)?

We implemented colour in the coverage maps and also clarified that this is effective coverage in the legend with the % mapping coverage stated for comparison.

Is there any publication related to the DeutEx software?

Thanks for the interest in the software, we now cite a manuscript, where it has previously been used.

Fig S16 – I think that the numbering here is shifted by 1. All proteins are equally numbered (based on Fig S1 alignment) up to position 298. Hence the numbering should also start with 220 like in the other maps. However this numbering is likely incorrect as it incorporates four initial amino acids from the expression vector (GPGS). In my point of view, the numbering should start on the fifth amino acid (Lys) in all maps.

Well spotted, we corrected the numbering. The linker residues are named in the methods but for convenience also named in the coverage maps now. The linker does however not affect overall numbering as this includes the N-terminal shell domain not present in our P domain constructs.

Does it make sense to show different conditions for peptide coverage in FigS12-16? Please explain! If the comparison is targeted the just peptides found in all relevant conditions are more appropriate dataset as the aim is anyway comparison and that needs same peptides in both (or all) conditions under comparison.

The coverage maps represent peptide coverage from individual HDX experiments for each strain. We have clarified this in the Figure legends, now stating e.g. “Sequence coverage is shown for two individual HDX-MS datasets with either B trisaccharide or fucose.”

What does it mean dimer wt vs. deam in Fig S15? Is that a comparison of two states or their difference?

We have rephrased to clarify that it refers to the peptides used in the comparison of the two states.